# From Correspondence to Actions: Human-Like Multi-Image Spatial Reasoning in Multi-modal Large Language Models

**Masanari Oi** [1]  **Koki Maeda** [1]  **Ryuto Koike** [1]  **Daisuke Oba** [1]  **Nakamasa Inoue** [1]  **Naoaki Okazaki** [1 2 3]

## Abstract

While multimodal large language models (MLLMs) have made substantial progress in single-image spatial reasoning, multi-image spatial reasoning, which requires integration of information from multiple viewpoints, remains challenging. Cognitive studies suggest that humans address such tasks through two mechanisms: *cross-view correspondence*, which identifies regions across different views that correspond to the same physical locations, and *stepwise viewpoint transformation*, which composes relative viewpoint changes sequentially. However, existing studies incorporate these mechanisms only partially and often implicitly, without explicit supervision for both. We propose Human-Aware Training for Cross-view correspondence and viewpoint cHange (HATCH), a training framework with two complementary objectives: (1) Patch-Level Spatial Alignment, which encourages patch representations to align across views for spatially corresponding regions, and (2) Action-then-Answer Reasoning, which requires the model to generate explicit viewpoint transition actions before predicting the final answer. Experiments on three benchmarks demonstrate that HATCH consistently outperforms baselines of comparable size by a clear margin and achieves competitive results against much larger models, while preserving single-image reasoning capabilities. Project page: https://stjohn2007.github.io/HATCH_project/

[1]Department of Computer Science, Institute of Science Tokyo, Japan [2]AIRC, National Institute of Advanced Industrial Science and Technology (AIST), Japan [3]LLMC, National Institute of Informatics (NII), Japan. Correspondence to: Masanari Oi <masanari.oi@nlp.comp.isct.ac.jp>.

*Proceedings of the 43rd International Conference on Machine Learning*, Seoul, South Korea. PMLR 306, 2026. Copyright 2026 by the author(s).

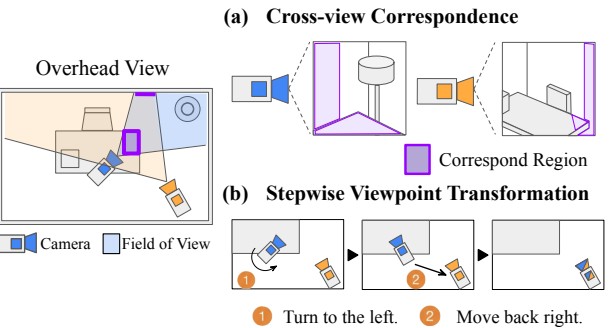

**(a) Cross-view Correspondence**

Overhead View

Camera | Field of View | Correspond Region

**(b) Stepwise Viewpoint Transformation**

1 Turn to the left.  2 Move back right.

*Figure 1.* Two cognitive mechanisms underlying multi-image spatial reasoning: (a) *cross-view correspondence*, identifying regions across views that correspond to the same physical locations; (b) *stepwise viewpoint transformation*, composing relative viewpoint changes (e.g., rotations) in a sequential manner.

## 1. Introduction

Multi-image spatial reasoning requires models to answer questions when the necessary evidence is distributed across multiple partial views of a physical scene (Wu et al., 2026; Gholami et al., 2026; Zhang et al., 2025; Wang et al., 2026; Yang et al., 2026). Formally, given a set of images from different viewpoints and a question, the model must produce an answer by integrating information across the views[1]. Unlike single-image settings, this task demands not only independent interpretation of each view but also the alignment and integration of partial observations across viewpoints (Yang et al., 2026). Despite substantial progress in single-image spatial reasoning for multimodal large language models (MLLMs) (Chen et al., 2024; Cheng et al., 2024; Cai et al., 2025; Song et al., 2025; Sun et al., 2025; Liu et al., 2025b; Chen et al., 2025a; Wu et al., 2025b; Li et al., 2025), current models often struggle to reliably aggregate information across multiple viewpoints for effective multi-image reasoning (Wang et al., 2026; Yang et al., 2026). This gap motivates a closer examination of what learning signals are missing for effective multi-image spatial reasoning.

Cognitive studies (Rock, 1983; Diwadkar & McNamara, 1997; Shepard & Metzler, 1971; Hegarty & Waller, 2004;

---

[1]In this work, we focus on *static* multi-view settings, where the scene geometry is consistent across views.

Wang & Spelke, 2000; Klatzky et al., 1998; Poth et al., 2015) provide key insights into this challenge: humans often reason across multiple viewpoints by (1) establishing spatial correspondences across views and (2) performing stepwise viewpoint transformations. As illustrated in Figure 1, *cross-view correspondence* allows humans to recognize regions in different views that correspond to the same physical locations or entities, despite appearance variations, occlusions, and partial overlaps. Complementarily, *stepwise viewpoint transformation* enables humans to take a sequence of actions to achieve viewpoint changes and to leverage these transformations for spatial reasoning.

Existing methods incorporate these cognitive insights only partially and often implicitly. Some studies primarily rely on large-scale fine-tuning with question–answer pairs to improve spatial reasoning performance, without explicitly modeling multi-image mechanisms (Daxberger et al., 2025; Ray et al., 2025). Other studies implicitly address cross-view correspondence by introducing supervision from 3D-specialized models or augmenting MLLMs with geometry-aware encoders (Wu et al., 2025a; Huang et al., 2025; Fan et al., 2026; Chen et al., 2026). Separately, viewpoint transformation has been explored through map-based or mental-map reasoning formulations (Ouyang et al., 2025; Wang et al., 2026), or via task-specific inference pipelines (Liu et al., 2025a; Lee et al., 2025). Overall, the joint and explicit incorporation of both cross-view correspondence and stepwise viewpoint transformation into a unified learning objective remains largely unexplored.

In this work, we propose **H**uman-**A**ware **T**raining for **C**ross-view correspondence and viewpoint c**H**ange (**HATCH**), a training framework for MLLMs that explicitly incorporates two complementary objectives, inspired by human spatial cognition. First, **Patch-Level Spatial Alignment** (PaStA) encourages cross-view correspondence by training the model to align patch features across views for spatially corresponding regions. Second, **Action-then-Answer Reasoning** (ActoR) promotes stepwise viewpoint transformation by generating intermediate actions for viewpoint transitions prior to answer prediction. In this stage, we optimize the model using GRPO (Shao et al., 2024), guided by two verifiable rewards: an answer reward that evaluates the final prediction and an action reward that supervises the generated viewpoint transitions. HATCH applies these two components sequentially: PaStA first teaches the model "how to look", perceiving spatial correspondence across views, and then ActoR teaches "how to move", reasoning through generation of viewpoint transition actions.

Experiments on three multi-image spatial reasoning benchmarks (Zhang et al., 2025; Wang et al., 2026; Yang et al., 2026) demonstrate the effectiveness of HATCH. Specifi-

cally, HATCH improves performance over the base model (Qwen2.5-VL-3B-Instruct) by 14.2% on average and consistently outperforms baselines with comparable parameter sizes by a clear margin. Ablation studies further show that jointly modeling PaStA and ActoR leads to stronger multi-image spatial reasoning and achieves the best performance among all variants. Moreover, HATCH attains strong and competitive performance on single-image spatial reasoning benchmarks, suggesting that our approach improves multi-image reasoning while remaining broadly effective in single-image settings.

Our contributions are summarized as follows:

- We introduce HATCH, a training framework for multi-image spatial reasoning that explicitly supervises both cross-view correspondence and stepwise viewpoint transformation, grounded in insights from human spatial cognition.

- We propose Patch-Level Spatial Alignment (PaStA), a geometry-supervised alignment objective that encourages the model to learn patch-level representations in which spatially corresponding regions across views are aligned in the feature space.

- We propose Action-then-Answer Reasoning (ActoR), which promotes stepwise viewpoint transformation by requiring the model to generate intermediate actions of viewpoint transitions prior to answer prediction, optimized via reinforcement learning with verifiable rewards.

- Experiments on three multi-image spatial reasoning benchmarks demonstrate that HATCH consistently outperforms baselines of comparable size and achieves competitive performance against substantially larger models. Ablation studies further show that both PaStA and ActoR are essential, and that HATCH also performs competitively on single-image spatial reasoning benchmarks.

## 2. Related Work

**Multi-Image Spatial Reasoning.** Multi-image spatial reasoning requires models to answer questions by integrating information from multiple views of a physical scene (Zhang et al., 2025; Wang et al., 2026; Yang et al., 2026). Existing approaches often address this challenge by introducing explicit spatial or 3D representations, including geometry-grounded models, 3D-aware reasoning frameworks, or learned spatial abstractions across views (Hu et al., 2026; Chen et al., 2026; Fan et al., 2026; Lee et al., 2025; Wang et al., 2026; Ouyang et al., 2025; Xu et al., 2026). While these methods enhance multi-image reasoning

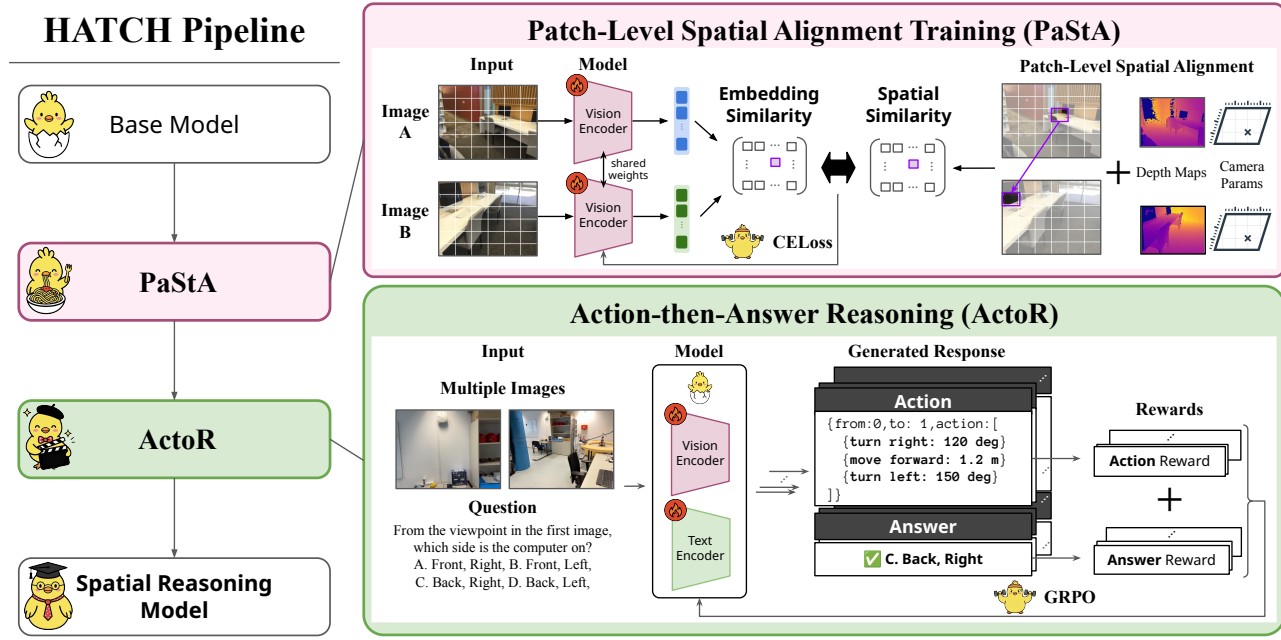

*Figure 2.* HATCH pipeline overview. HATCH consists of two components: (i) Patch-Level Spatial Alignment (PaStA) to learn cross-view correspondence, (ii) Action-then-Answer Reasoning (ActoR) to perform stepwise viewpoint transformation via explicit actions.

through architectural design or implicit supervision, they do not explicitly supervise both cross-view correspondence and stepwise viewpoint transformation within a unified learning objective. In contrast, HATCH directly incorporates both mechanisms through explicit feature-level alignment and action-based viewpoint transformation training.

**Feature-Level Alignment.** Feature-level alignment between vision and language is widely used to learn transferable visual representations and enable zero-shot generalization (Radford et al., 2021; Jia et al., 2021; Zhai et al., 2022; 2023). Beyond global image-text alignment, prior work explores fine-grained objectives that align regions or objects with language, improving locality and compositionality (Zhong et al., 2022; Li et al., 2022; Gu et al., 2022). In parallel, dense self-supervised objectives enforce pixel- or patch-level consistency to preserve spatial structure, yielding representations better suited for localized reasoning (Wang et al., 2021; Xie et al., 2021; Caron et al., 2021; Zhou et al., 2022). More closely related to cross-view correspondence, CroCo (Weinzaepfel et al., 2022) learns representations from paired views of the same scene via cross-view completion, encouraging geometric consistency across viewpoints. However, these methods are primarily designed for representation learning and do not explicitly support multi-image spatial reasoning. Building on these ideas, Patch-Level Spatial Alignment (PaStA) adapts feature-level alignment to multi-image spatial reasoning by using geometry supervision to align patch features corresponding to the same 3D locations across views.

**Intermediate Reasoning and Verifiable Rewards.** Introducing intermediate reasoning steps improves performance on complex tasks, ranging from chain-of-thought prompting to action-based and multi-path reasoning frameworks (Wei et al., 2022; Kojima et al., 2022; Yao et al., 2023b;a). More recently, this paradigm has been combined with *verifiable rewards*, where intermediate or final outputs are evaluated by automated verifiers, enabling reinforcement learning with reduced reliance on human preference supervision. DeepSeekMath introduces Group Relative Policy Optimization (GRPO) as an efficient training algorithm for such settings (Shao et al., 2024). This approach has also been extended to spatial reasoning (Liao et al., 2025; Ouyang et al., 2025; Wang et al., 2026; Li et al., 2025). MindCube (Wang et al., 2026) applies GRPO by treating an explicitly constructed mental map as the intermediate reasoning target, while SpatialLadder (Li et al., 2025) strengthens VLM spatial reasoning through progressive training with task-specific verifiable rewards. While these methods demonstrate the effectiveness of verifiable reward learning for spatial reasoning, they primarily operate over abstract intermediate representations. In contrast, ActoR focuses on *stepwise viewpoint-transition actions* as intermediate steps and trains them using both geometry-based and answer-based verifiable rewards, directly aligning intermediate supervision with the viewpoint transformations.

## 3. Methodology

### 3.1. Problem Setup

An input to multi-image spatial reasoning consists of a set of $N$ images $\mathcal{I} = \{I_1, I_2, \ldots, I_N\}$ capturing the same physical scene from different viewpoints, and a natural language question $Q$ whose answer requires spatial reasoning across views. At both training and inference, the model receives only $(\mathcal{I}, \mathbf{Q})$ as input. During training, we additionally assume access to camera intrinsics, camera poses, and depth maps for each image. These geometric signals are used solely to construct supervision for PaStA and verifiable rewards for ActoR; they are never provided as model inputs.

In this work, we focus on *static* scenes, where the scene geometry is consistent across views, so that a single 3D structure underlies all viewpoints. This setting covers many practical scenarios, such as multiple surveillance cameras or multi-robot systems observing the same environment from different viewpoints. This assumption underlies the geometric supervision used in PaStA (described below), which relies on consistent 3D correspondences across views.

### 3.2. Overview of HATCH

As illustrated in Figure 2, HATCH addresses the two cognitive mechanisms identified in the introduction through complementary training objectives. Patch-Level Spatial Alignment (PaStA) targets cross-view correspondence by encouraging patch representations to align across views when they correspond to the same 3D locations. Action-then-Answer Reasoning (ActoR) targets stepwise viewpoint transformation by requiring the model to generate intermediate viewpoint transition actions prior to predicting the final answer. These components are applied sequentially during training. We first apply PaStA to update only the image encoder while keeping the language model frozen, teaching the model *how to look* across views without entangling correspondence learning with language generation. We then train the model with ActoR, teaching the model *how to move* through explicit viewpoint transitions.

### 3.3. Patch-Level Spatial Alignment (PaStA)

**Motivation.** A core failure mode in multi-image settings is that spatially corresponding regions across viewpoints do not map to consistent representations, forcing the model to implicitly infer alignment during reasoning (Yang et al., 2026). PaStA addresses this issue by leveraging training-time geometric information to construct patch-level correspondence targets and explicitly supervise the image encoder to align representations across views.

**Patch-Level Spatial Overlap.** Given a pair of images $(X, Y)$ observing the same scene, we divide each image

into an $n \times n$ grid of patches and compute a patch-to-patch spatial overlap matrix across views using camera intrinsics, camera poses, and depth maps (available only at training time). For each patch $i$ in image $X$, we reconstruct 3D points from pixels within the patch using the depth map and project them into image $Y$ via the world coordinate system. A projected point is considered geometrically consistent if its projected depth matches the depth value in image $Y$ within a threshold $t$. We define a directional overlap matrix $M_{X \to Y} \in \mathbb{R}^{(n^2) \times (n^2)}$, where each element $M_{X \to Y}[i, j]$ denotes the fraction of pixels in patch $i$ of $X$ whose reconstructed 3D points project consistently into patch $j$ of $Y$. To obtain a symmetric measure of spatial correspondence, we compute

$$S = \frac{1}{2}\left(M_{X \to Y} + M_{Y \to X}^{\top}\right), \tag{1}$$

where $S \in \mathbb{R}^{(n^2) \times (n^2)}$ where each element $S[i, j] \in [0, 1]$ quantifies the spatial correspondence between patch $i$ in image $X$ and patch $j$ in image $Y$.

**Correspondence Alignment Objective Formulation.** Let $E(\cdot)$ denote the image encoder. For each image, we obtain patch-level features by partitioning the encoder output into an $n \times n$ grid, yielding feature vectors $\{\mathbf{e}_i\}_{i=1}^{n^2}$. Given a pair of images $(X, Y)$, we define a geometry-derived target correspondence distribution from patch $i$ in image $X$ to patches in image $Y$ as

$$p(j \mid i) = \text{softmax}_j\left(\frac{S[i, :]}{\tau_1}\right), \tag{2}$$

where $\tau_1$ is a temperature parameter and the softmax is taken over all patches $j$ in image $Y$. Similarly, the model-predicted correspondence distribution is defined based on cosine similarity between patch features:

$$q(j \mid i) = \text{softmax}_j\left(\frac{\cos(\mathbf{e}_i^X, \mathbf{e}_:^Y)}{\tau_2}\right), \tag{3}$$

where $\tau_2$ is a temperature parameter, and $\mathbf{e}_i^X$ (resp. $\mathbf{e}_j^Y$) denotes the feature vector of patch $i$ in image $X$ (resp. patch $j$ in image $Y$). Intuitively, $p(\cdot \mid i)$ provides a soft correspondence target that tolerates partial overlap and occlusion, while $q(\cdot \mid i)$ reflects the encoder's similarity-induced matching.

We minimize the cross-entropy between the target and predicted correspondence distributions in both directions:

$$\mathcal{L}_{\text{CL}} = \frac{1}{|\mathcal{P}|} \sum_{(X,Y) \in \mathcal{P}} \left(\mathcal{L}_{X \to Y} + \mathcal{L}_{Y \to X}\right), \tag{4}$$

$$\mathcal{L}_{X \to Y} = \frac{1}{n^2} \sum_{i=1}^{n^2} \text{CE}\left(p(\cdot \mid i), q(\cdot \mid i)\right). \tag{5}$$

Here, $\mathcal{P}$ denotes the set of unordered image pairs $(X, Y)$ with $X \neq Y$ used for training. During PaStA training,

only the image encoder parameters are updated, while the language model remains frozen.

## 3.4. Action-then-Answer Reasoning (ActoR)

**Motivation.** Even when spatially corresponding regions are well aligned in the representation space, answering multi-image questions often requires composing viewpoint changes to synthesize evidence across views (e.g., inferring an object relation that becomes apparent only after a viewpoint shift). Rather than leaving such composition implicit in the reasoning process, ActoR introduces explicit viewpoint-transition actions as an intermediate representation. By making viewpoint changes explicit, this design encourages structured and interpretable reasoning, analogous to chain-of-thought prompting in language models (Wei et al., 2022; Kojima et al., 2022).

**Reasoning Formulation.** Given images $\mathcal{I} = \{I_1, \ldots, I_N\}$ and a question $Q$, ActoR formulates reasoning by explicitly generating viewpoint-transition actions, followed by answer prediction conditioned on those actions. The output takes the form:

$$\texttt{<action>}\ \mathcal{A}\ \texttt{</action>}\ \texttt{<answer>}\ a\ \texttt{</answer>}, \tag{6}$$

where $\mathcal{A}$ is a sequence of viewpoint-transition actions in JSON and $a$ is the final answer. We generate actions for all unordered image pairs $(i, j)$ with $i < j$:

$$\mathcal{A} = \{(i, j, \mathbf{a}_{i \to j}) \mid 1 \le i < j \le N\}, \tag{7}$$

where each $\mathbf{a}_{i \to j}$ consists of a sequence of atomic camera operations drawn from a fixed action vocabulary, including `turn_left/right_deg`, `turn_up/down_deg`, `move_forward_m`, and `move_up/down_m`. Refer to Figure 2 for an illustration of the structure of the action. A concrete example of the action JSON format is provided in the Appendix B.

**Cold-Start SFT.** Before applying reinforcement learning, we perform a cold-start supervised fine-tuning (SFT) phase to familiarize the model with the Action-then-Answer output format (DeepSeek-AI, 2025; Li et al., 2025). Teacher action sequences are constructed offline using relative camera poses by decomposing the relative transformations into rotations and translations. The goal of this stage is simply to establish the Action-then-Answer generation structure before reinforcement learning, rather than improving task performance.

**Reinforcement Learning with Verifiable Rewards.** After cold-start SFT, we refine the model using Group Relative Policy Optimization (GRPO) (Shao et al., 2024) to improve both the geometric accuracy of generated actions and the correctness of final answers.

The total reward is a weighted sum of three components:

$$R = \lambda_1 R_{\text{act-acc}} + \lambda_2 R_{\text{ans-acc}} + \lambda_3 R_{\text{format}}, \tag{8}$$

where $R_{\text{act-acc}}$ evaluates the geometric accuracy of the predicted viewpoint-transition actions, $R_{\text{ans-acc}}$ measures the correctness of the final answer, and $R_{\text{format}}$ verifies whether the output follows the prescribed Action-then-Answer format. The format reward is binary, while the accuracy rewards provide continuous feedback based on geometric consistency (for actions) and task-specific metrics (for answers). Detailed definitions of each reward are in the Appendix C.

**Inference.** At inference time, the model receives only the input images $\mathcal{I}$ and the question $Q$, and generates outputs in the same Action-then-Answer format. The generated actions serve as an explicit intermediate reasoning step that precedes answer generation; no camera intrinsics, poses, or depth maps are required.

**Training Recipe.** We adopt a staged training recipe to progressively equip the model with the capabilities required for multi-image spatial reasoning. We first apply PaStA to tune the image encoder for cross-view correspondence while keeping the language model frozen. Next, we perform standard supervised fine-tuning on question–answer pairs (without action annotations) to familiarize the full multi-modal model with answer generation for the target task. We then apply ActoR (cold-start SFT and GRPO) to improve both action generation for viewpoint transition and final answer prediction.

**Discussion on Computational Cost.** Although HATCH adopts a multi-stage training recipe and incorporates reinforcement learning over action sequences, its overall computational overhead remains modest. The additional stages introduced by PaStA and the cold-start SFT incur limited cost: PaStA updates only the image encoder, making it significantly lighter than full multimodal training, while the cold-start SFT uses a small fraction of data solely to establish the Action-then-Answer format. Moreover, compared to standard GRPO applied to free-form language reasoning, ActoR introduces only a lightweight additional computation for action rewards, which are based on simple geometric comparisons between predicted and target motion vectors.

## 4. Experiments

We evaluate HATCH on three multi-image spatial reasoning benchmarks: SPAR-Bench-MV[2] (Zhang et al., 2025), MindCube-Tiny (Wang et al., 2026), and MMSI-Bench (Yang et al., 2026). Details of evaluation benchmarks are in the Appendix E.

---

[2]We extract multi-image samples from SPAR-Bench and refer to the resulting dataset as SPAR-Bench-MV.

*Table 1.* Main results. **Bold** and Underlined numbers indicate the best performance among Qwen2.5-VL-3B-based models and all open-weight models, respectively. Results of proprietary models are shown for reference and are excluded from open-weight comparisons. *Gray italic* numbers denote evaluation on SPAR-Bench-Tiny-MV (a subset of SPAR-Bench-MV). Abbreviations: Mid. = Middle; Aro. = Around; Rot. = Rotation; Amo. = Among; Pos. = Position; Att. = Attribute; Mot. = Motion; MSR = Multi-step Reasoning.

| Model | SPAR-Bench-MV | | | | MindCube-Tiny | | | | MMSI-Bench | | | | | Overall |
|---|---|---|---|---|---|---|---|---|---|---|---|---|---|---|
| | Low | Mid. | High | Avg. | Aro. | Rot. | Amo. | Avg. | Pos. | Att. | Mot. | MSR | Avg. | |
| *Proprietary* | | | | | | | | | | | | | | |
| GPT-4.1 | 38.2 | 37.0 | 43.1 | 39.9 | 59.6 | 49.5 | 47.2 | 50.6 | 29.3 | 33.8 | 38.7 | 31.3 | 31.7 | 40.7 |
| GPT-5.2 | *43.7* | *41.5* | *66.4* | *52.6* | 69.2 | 95.5 | 41.5 | 58.4 | 42.9 | 47.6 | 36.0 | 40.4 | 42.0 | 51.0 |
| Gemini-3-Pro | *40.6* | *21.7* | *58.0* | *43.1* | 51.6 | 54.5 | 46.0 | 49.0 | 42.9 | 50.0 | 38.0 | 40.9 | 42.7 | 44.9 |
| *Open-Weight* | | | | | | | | | | | | | | |
| InternVL-2.5-4B | 30.3 | 30.1 | 38.5 | 33.6 | 57.6 | 33.5 | 45.2 | 45.9 | 28.7 | 26.2 | 20.7 | 28.3 | 27.1 | 35.5 |
| InternVL-2.5-8B | 31.8 | 30.8 | 48.2 | 38.3 | 25.6 | 37.0 | 20.3 | 24.8 | 31.0 | 30.0 | 20.7 | 23.2 | 27.8 | 30.3 |
| LLaVA-OneVision-4B | 23.0 | 28.0 | 40.1 | 31.5 | 50.0 | 33.5 | 39.2 | 40.7 | 30.8 | 26.9 | 26.0 | 26.8 | 28.8 | 33.7 |
| Qwen2.5-VL-32B | 25.6 | 29.0 | 45.4 | 34.7 | 43.6 | 41.5 | 38.2 | 40.0 | 25.9 | 26.9 | 28.7 | 27.3 | 26.7 | 33.8 |
| Qwen2.5-VL-72B | 26.9 | 32.7 | 43.6 | 35.4 | 44.8 | 42.0 | 41.2 | 42.2 | 33.3 | 38.5 | 24.7 | 28.8 | 31.8 | 36.5 |
| *Qwen2.5-VL-7B Based* | | | | | | | | | | | | | | |
| Qwen2.5-VL-7B | 16.9 | 29.4 | 40.7 | 30.1 | 38.0 | 33.5 | 32.7 | 34.1 | 28.4 | 21.5 | 29.3 | 25.8 | 27.1 | 30.4 |
| SpaceR-7B | 31.6 | 32.3 | 49.2 | 39.1 | 31.6 | 31.5 | 28.7 | 29.9 | 30.3 | 25.4 | 22.0 | 21.7 | 26.7 | 31.9 |
| Video-R1 | 29.4 | 30.0 | 46.2 | 36.5 | 44.0 | 30.0 | 38.5 | 38.2 | 30.5 | 33.8 | 18.7 | 24.7 | 28.0 | 34.2 |
| *Qwen2.5-VL-3B Based* | | | | | | | | | | | | | | |
| Qwen2.5-VL-3B | 13.4 | 26.2 | 32.6 | 24.9 | 46.4 | **34.5** | 35.3 | 37.8 | 27.8 | 19.2 | 23.3 | **25.8** | 25.6 | 29.4 |
| Spatial-MLLM-4B | 22.3 | 32.4 | 35.2 | 30.4 | 52.0 | 29.0 | 34.5 | 37.6 | 28.0 | 21.5 | 14.7 | 22.7 | 24.1 | 30.7 |
| SpatialLadder-3B | 26.2 | 33.5 | 44.5 | 35.8 | 56.4 | 31.5 | 47.8 | 46.8 | 24.1 | **23.1** | **26.7** | 20.2 | 23.6 | 35.4 |
| HATCH (Proposed) | **41.3** | **47.4** | **67.1** | **53.6** | **65.6** | 31.0 | **50.2** | **50.2** | 30.5 | 20.8 | 23.3 | 24.7 | **27.0** | **43.6** |

## 4.1. Experimental Settings

**Baselines.** We employ baselines from three categories and compare with HATCH.

*Proprietary models.* We evaluate GPT-4.1 (OpenAI, 2024), GPT-5.2 (OpenAI, 2025), and Gemini-3-Pro (Google, 2025) via their respective APIs. Because of API cost constraints for GPT-5.2 and Gemini-3-Pro, we evaluate on SPAR-Bench-Tiny-MV, an official subset of SPAR-Bench-MV, and mark the corresponding table entries in gray.

*Open-weight MLLMs.* We include five open-weight models: InternVL-2.5-4B/8B (Chen et al., 2025b), LLaVA-OneVision-4B (An et al., 2025), and Qwen2.5-VL-32B/72B-Instruct (Qwen Team, 2025).

*Spatial reasoning models.* We further compare against models explicitly trained for spatial reasoning, including Qwen2.5-VL-7B-based models (SpaceR-7B (Ouyang et al., 2025) and Video-R1 (Feng et al., 2025)) and Qwen2.5-VL-3B-based models (Spatial-MLLM-4B (Wu et al., 2025a) and SpatialLadder-3B (Li et al., 2025)), where the latter serve as the primary baselines for HATCH.

**HATCH.** We construct the training data for HATCH from SPAR-7M (Zhang et al., 2025) by selecting samples with multiple images and randomly sampling 10,000 instances for training. For each instance, we derive patch-level correspondence signals and viewpoint-transition action se-

quences to train PaStA and ActoR, respectively. We adopt Qwen2.5-VL-3B (Qwen Team, 2025) as the base model across all experiments. For the cold-start SFT stage of ActoR, we use a 10% subset of the training data, whereas all other stages use the full dataset. Additional implementation details are provided in the Appendix D.

## 4.2. Main Results

**Improvements over the Baselines.** As summarized in Table 1, HATCH substantially improves upon the base model Qwen2.5-VL-3B across all benchmarks, achieving 53.6% (+28.7 points) on SPAR-Bench-MV, 50.2% (+12.4 points) on MindCube-Tiny, and 27.0% (+1.4 points) on MMSI-Bench. In addition, HATCH consistently outperforms prior spatial reasoning methods built on the same backbone, attaining the highest average performance among all Qwen2.5-VL-3B-based models on all three benchmarks, thereby demonstrating the effectiveness of our approach. Although gains on MMSI-Bench are modest in categories such as Attribute and Motion, most evaluated models perform at or below chance level (25.0%) in these categories. This makes it difficult to interpret the difference reliably and suggests the remaining challenge for the current approaches.

**Comparison with Larger Open-Weight Models.** Despite using Qwen-2.5-VL-3B backbone, HATCH outperforms larger open-weight models, 7B-based spatial reasoning mod-

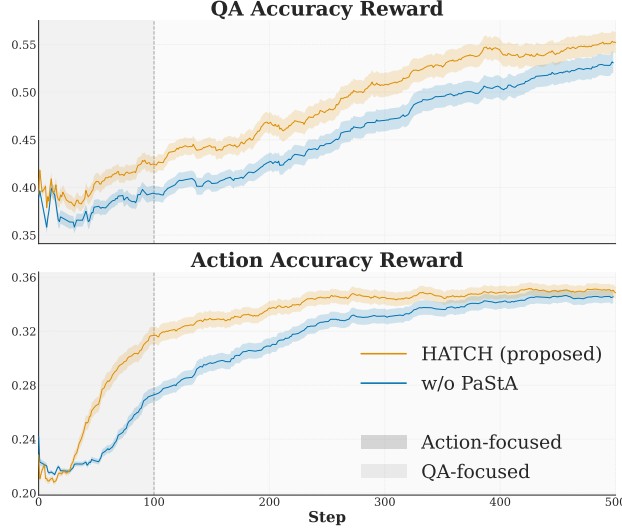

*Figure 3.* Training dynamics during GRPO training of ActoR. Action and QA accuracy rewards are shown for HATCH (yellow) and an ablated variant without PaStA (blue).

*Table 2.* Component ablation on SPAR-Bench-MV. "w/o both" indicates only cold-start SFT with QA pairs are conducted. Subscripts denote the change in accuracy relative to HATCH.

| Method | SPAR-Bench-MV | | | |
|---|---|---|---|---|
| | Low | Middle | High | Avg. |
| HATCH | **41.3** | **47.4** | 67.1 | **53.6** |
| *w/o* PaStA | $39.3_{-2.0}$ | $43.5_{-3.9}$ | $67.4_{+0.3}$ | $52.0_{-1.6}$ |
| *w/o* ActoR | $35.4_{-5.9}$ | $45.8_{-1.6}$ | $66.5_{-0.6}$ | $51.1_{-2.5}$ |
| *w/o* both | $36.9_{-4.4}$ | $44.8_{-2.6}$ | $67.1_{0.0}$ | $51.5_{-2.1}$ |

els and 32B/72B, on SPAR-Bench-MV and MindCube-Tiny. HATCH remains competitive on MMSI-Bench, indicating that the observed gains stem from improved reasoning structure rather than increased model capacity.

**Comparison with Proprietary Models.** On SPAR-Bench-MV and MindCube-Tiny, HATCH achieves competitive performance with state-of-the-art proprietary models. In particular, HATCH (53.6%) matches GPT-5.2 (52.6%) on SPAR-Bench-MV and approaches its performance on MindCube-Tiny (50.2% vs. 58.4%). On MMSI-Bench, a larger gap remains, with GPT-5.2 achieving 42.0% compared to 27.0% for HATCH. This gap suggests that while HATCH effectively addresses correspondence- and viewpoint-based multi-image reasoning, further enhancements are needed to handle more diverse reasoning requirements.

### 4.3. Analysis

**Training Dynamics.** Figure 3 plots the QA accuracy reward and action accuracy reward during GRPO training of ActoR, comparing HATCH (yellow) with an ablated variant without PaStA (blue). Training exhibits a clear two-phase

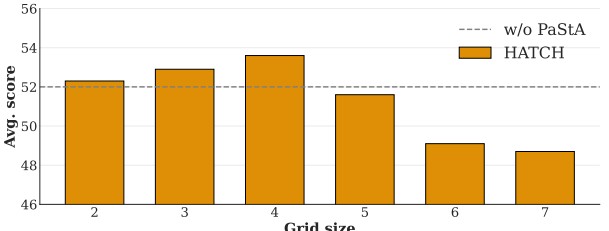

*Figure 4.* Grid resolution analysis for PaStA. SPAR-Bench-MV average accuracy improves up to $n = 4$ and drops for $n \geq 5$, indicating that overly fine grids hurt correspondence learning.

*Table 3.* Comparison of reasoning modalities in ActoR on SPAR-Bench-MV across categories.

| Reasoning modality | SPAR-Bench-MV | | | |
|---|---|---|---|---|
| | Low | Middle | High | Avg. |
| Natural Language | 40.7 | 46.4 | 66.1 | 52.8 |
| Sequential Action (HATCH) | **41.3** | **47.4** | 67.1 | **53.6** |

pattern: action reward improves early, and QA reward increases later as the learned viewpoint-transition actions are exploited for answer prediction. This behavior aligns with the design of ActoR, where explicit viewpoint-transition actions serve as an intermediate reasoning step. By contrast, removing PaStA consistently degrades both action and QA rewards throughout training, indicating that correspondence-aligned representations provide a stronger foundation for accurate action generation and downstream reasoning.

**Component Ablation.** Table 2 reports the results of ablating individual components on SPAR-Bench-MV. Here, "*w/o both*" denotes supervised fine-tuning using only question–answer pairs, without PaStA or ActoR. Ablating either PaStA, ActoR, or both consistently degrades performance overall, with the two components contributing in complementary ways. Removing PaStA leads to a pronounced drop in the Middle category, which emphasizes cross-view reasoning under viewpoint changes, indicating its role in learning view-consistent representations. In contrast, removing ActoR primarily affects the Low category, dominated by depth and distance estimation, suggesting that explicit viewpoint-transition actions facilitate geometric inference. Overall, PaStA and ActoR address distinct but complementary aspects of multi-image spatial reasoning.

**Effect of Grid Resolution in PaStA.** We analyze the effect of the grid resolution $n$ in PaStA by varying the number of patches. Figure 4 plots the average accuracy on SPAR-Bench-MV as $n$ increases. Performance improves as the grid resolution increases from $n = 2$ to 4, indicating that finer spatial partitioning enables more precise correspondence learning. However, performance drops for $n \geq 5$, even falling below the variant without PaStA. These results

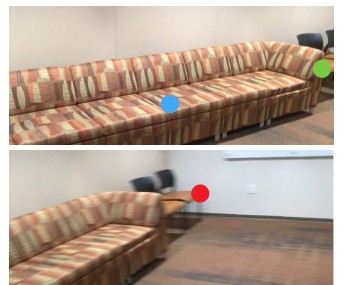

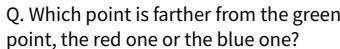

📝 **Reasoning by Natural Language**

**\<think\>**

*We need to compare the distance of couch with the blue point and the chair with the red point.*
*1. The couch is centrally positioned in the image and appears closer to the chair with the green point.*
*2. The chair with red point is located further away from both the blue and green point.*
*Given these observations, the chair with red point*
**\</think\>**
**\<answer\>** ***The red point.*** **\</answer\>**

🎓 **HATCH: Reasoning by Action (Proposed)**

**\<action\>**

```
{from: 0, to: 1, action: [
  {turn up: 15 deg}, {turn right: 25 deg},
  {move forward: 0.4 m}, {move up: 0.1 m},
  {turn left: 25 deg}, {turn down: 15 deg}
]}
```

**\</action\>**

**\<answer\>** ***The blue point.*** **\</answer\>**

Q. Which point is farther from the green point, the red one or the blue one?

*Figure 5.* Qualitative success and failure cases for different reasoning modalities. Compared with natural language reasoning, action-based reasoning (HATCH) yields explicit, quantitative camera operations that more directly support correct multi-image spatial inference.

*Table 4.* Performance on single-image spatial reasoning benchmarks (SPAR-Bench-SI and CV-Bench).

| Model | SPAR-Bench-SI | | | CV-Bench | | |
|---|---|---|---|---|---|---|
| | Low | High | Avg. | 2D | 3D | Avg. |
| *Proprietary* | | | | | | |
| GPT-4.1 | 46.5 | 50.1 | 48.2 | 75.0 | 86.9 | 80.9 |
| GPT-5.2 | *54.3* | *66.0* | *60.2* | 77.9 | 90.4 | 84.2 |
| Gemini-3-Pro | *52.5* | *54.5* | *53.5* | 80.7 | 91.2 | 85.9 |
| *Open-Weight* | | | | | | |
| InternVL-2.5-4B | 32.1 | 33.7 | 32.9 | 73.2 | 74.6 | 73.9 |
| InternVL-2.5-8B | 32.9 | 44.1 | 38.3 | 74.0 | 79.5 | 76.8 |
| LLaVA-OneVision-4B | 27.6 | 41.4 | 34.2 | 70.8 | 79.2 | 75.0 |
| Qwen2.5-VL-32B | 35.6 | 49.3 | 42.2 | 76.8 | 84.0 | 80.4 |
| Qwen2.5-VL-72B | 32.7 | 48.0 | 40.1 | 77.1 | 84.2 | 80.7 |
| *Qwen2.5-VL-7B Based* | | | | | | |
| Qwen2.5-VL-7B | 18.7 | 45.2 | 31.4 | 74.8 | 83.3 | 79.1 |
| SpaceR-7B | 33.6 | 46.3 | 39.7 | 71.9 | 81.1 | 76.4 |
| Video-R1 | 33.8 | 42.7 | 38.1 | 71.7 | 74.9 | 73.3 |
| *Qwen2.5-VL-3B Based* | | | | | | |
| Qwen2.5-VL-3B | 16.0 | 32.5 | 23.9 | 69.3 | 72.2 | 70.7 |
| Spatial-MLLM-4B | 24.3 | 37.6 | 30.7 | 65.7 | 69.5 | 67.6 |
| SpatialLadder-3B | 25.7 | 39.8 | 32.5 | **72.2** | 74.6 | 73.4 |
| HATCH | **41.1** | **49.7** | **45.2** | 70.5 | **78.4** | **74.5** |

suggest that overly fine grids fragment visual regions beyond reliable correspondence matching, making $n = 4$ a balance point between spatial resolution and training stability.

**Effect of Reasoning Modality in ActoR.** To disentangle the effect of action-based reasoning from GRPO training itself, we compare HATCH with a variant that generates free-form natural-language chain-of-thought reasoning before the answer (Feng et al., 2025; Li et al., 2025). Both variants are trained with the same GRPO framework, dataset, and initialization, but with different reasoning modality.

As shown in Table 3, action-based reasoning (HATCH) consistently outperforms natural language reasoning across all categories on SPAR-Bench-MV. Qualitative examples in

Figure 5 further illustrate this difference. Natural language reasoning often produces vague or imprecise spatial descriptions (e.g., "appears closer to the chair"), which can lead to incorrect answers. In contrast, action-based reasoning generates explicit and quantitative camera operations (e.g., "move forward: 0.4 m"), providing verifiable geometric cues that directly support accurate answer prediction.

While the two modalities are not mutually exclusive, our results indicate that action-based reasoning is more effective for multi-image spatial reasoning under the current setting. Exploring hybrid strategies that combine geometric actions with verbal reasoning is left for future work.

**4.4. Performance on Single-Image Spatial Reasoning**

Although HATCH is designed for multi-image spatial reasoning, we evaluate whether the proposed training framework generalizes to single-image regime. We test HATCH on two single-image spatial reasoning benchmarks: SPAR-Bench-SI[3] and CV-Bench (Tong et al., 2024). In this setting, HATCH is prompted to answer directly without generating viewpoint-transition actions.

As shown in Table 4, HATCH substantially improves over its base model Qwen2.5-VL-3B, raising the average accuracy from 23.9% to 45.2% (+21.3 points) on SPAR-Bench-SI and from 70.7% to 74.5% (+3.8 points) on CV-Bench. This represents the best performance among all Qwen2.5-VL-3B-based models on both benchmarks. On SPAR-Bench-SI, HATCH also outperforms Qwen2.5-VL-7B-based spatial reasoning models and several larger open-weight models, including Qwen2.5-VL-72B. On CV-Bench, HATCH remains competitive with other open-weight models, though a gap persists compared to proprietary models. Overall, these results indicate that the spatial reasoning abilities learned through PaStA and ActoR generalize beyond multi-image reasoning.

---

[3]We extract single-image samples from SPAR-Bench and refer to the resulting dataset as SPAR-Bench-SI.

## 5. Conclusion

This work addressed multi-image spatial reasoning in multimodal large language models, which required integrating information across multiple views of the same environment. We proposed HATCH, a training framework inspired by human spatial cognition that combined representation-level correspondence learning (PaStA) with an Action-then-Answer reasoning formulation (ActoR). Experiments on the three benchmarks showed that HATCH substantially improved multi-image spatial reasoning over the strong baselines while maintaining competitive performance on single-image tasks. Ablation studies further demonstrated that correspondence learning and structured action-based reasoning played complementary roles in enabling effective cross-view spatial understanding.

HATCH also has several limitations. It relies on geometric annotations (camera intrinsics, poses, and depth maps) during training, limiting its applicability to data where such metadata is available. It also assumes a static scene whose geometry is consistent across views, leaving temporal inconsistencies such as moving objects beyond its scope. We discuss these limitations in detail in Appendix F.

Future directions include reducing the dependence on geometric annotations by leveraging automatically estimated geometry and extending HATCH to handle dynamic scenes with temporal inconsistencies. Beyond addressing these limitations, we also plan to explore alternative correspondence supervision strategies, such as directly supervising attention mechanisms, and adaptive action-selection schemes to further reduce inference-time computation.

## Impact Statement

This paper presents work whose goal is to advance the field of Machine Learning. There are many potential societal consequences of our work, none which we feel must be specifically highlighted here.

## Acknowledgements

These research results were obtained from the commissioned research (No.22501) by National Institute of Information and Communications Technology (NICT), Japan. This work was partially supported by JSPS KAKENHI Grant Number 25H01137 and JST K Program Japan Grant Number JPMJKP24C3. This study was carried out using the TSUBAME4.0 supercomputer at Institute of Science Tokyo.

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

## A. Notation

We summarize the notation used throughout the paper in Table 5.

*Table 5.* Notation Table.

| Symbol | Meanings |
| --- | --- |
| $N$ | Number of input images (views). |
| $\mathcal{I} = \{I_1, \ldots, I_N\}$ | Set of input images capturing the same environment from different viewpoints. |
| $I_i$ | The $i$-th input image in $\mathcal{I}$. |
| $Q$ | Natural language question requiring spatial understanding across views. |
| $(X, Y)$ | A pair of images (views) observing the same scene, used for correspondence supervision. |
| $n$ | Grid resolution per side when dividing an image into patches; total patches per image is $n^2$. |
| $i, j$ | Patch indices (over the $n \times n$ patch grid). |
| $t$ | Geometric consistency threshold for depth agreement when projecting 3D points across views. |
| $M_{X \to Y}[i, j]$ | Directional overlap: fraction of valid points from patch $i$ in $X$ that project consistently into patch $j$ in $Y$. |
| $S$ | Symmetric patch correspondence matrix, $S = \frac{1}{2}(M_{X \to Y} + M_{Y \to X}^{\top})$, with $S[i, j] \in [0, 1]$. |
| $E(\cdot)$ | Vision encoder. |
| $\mathbf{e}_i$ | Feature vector for patch $i$ (from partitioning the encoder output into an $n \times n$ grid). |
| $\mathbf{e}_i^X, \mathbf{e}_j^Y$ | Patch features from image $X$ / image $Y$, used to compute cross-view similarity. |
| $p(j \mid i)$ | Target correspondence distribution from patch $i$ to patches $j$, derived from $S$ via softmax with temperature $\tau_1$. |
| $q(j \mid i)$ | Predicted correspondence distribution from patch features via softmax over cos-similarity with temperature $\tau_2$. |
| $\tau_1, \tau_2$ | Temperature parameters for the target / predicted correspondence softmax distributions. |
| $\mathcal{P}$ | Set of training image pairs $(X, Y)$. |
| $\mathrm{CE}(\cdot, \cdot)$ | Cross-entropy between two distributions. |
| $\mathcal{L}_{X \to Y}$ | Directional correspondence loss from $X$ to $Y$ (average cross-entropy over patches). |
| $\mathcal{L}_{\mathrm{CL}}$ | Overall correspondence-learning loss, averaging $\mathcal{L}_{X \to Y} + \mathcal{L}_{Y \to X}$ over pairs in $\mathcal{P}$. |
| $\mathcal{A}$ | Set of generated action annotations over ordered image pairs for structured reasoning. |
| $(i, j, \mathbf{a}_{i \to j})$ | Tuple denoting an ordered image pair with $i < j$ and its associated viewpoint-transition action sequence. |
| $\mathbf{a}_{i \to j}$ | Interpretable camera-operation sequence (rotations/translations with continuous parameters) moving from view $i$ to view $j$. |
| $a$ | Final predicted answer text. |
| $R$ | Scalar reinforcement-learning reward for a generated output. |
| $\lambda_1, \lambda_2, \lambda_3$ | Weights for reward components in $R$. |
| $R_{\mathrm{act\text{-}acc}}$ | Reward for geometric accuracy of predicted action sequences. |
| $R_{\mathrm{ans\text{-}acc}}$ | Reward for final answer correctness. |
| $R_{\mathrm{format}}$ | Reward for answer format validity. |

## B. Action Output

This section provides a concrete example of the action output format used in Action-then-Answer Reasoning (ActoR), as referenced in Section 3.4.

### B.1. Action Output Schema

Given $N$ input images, the model predicts relative camera motions for all ordered image pairs $(i, j)$. Each action sequence represents a discretized camera trajectory transforming the viewpoint of image $i$ to that of image $j$.

### B.2. JSON-formatted Example

For example, given three input images, the model generates actions for all image pairs in the following format:

```
[
  {
    "from": 0,
    "to": 1,
    "action": [
      {"turn_right_deg": 30},
      {"turn_up_deg": 15},
```

```
      {"move_forward_m": 1.4},
      {"move_up_m": 0.3}
    ]
  },
  {
    "from": 0,
    "to": 2,
    "action": [...]
  },
  {
    "from": 1,
    "to": 2,
    "action": [...]
  }
]
```

Each atomic action corresponds to a primitive camera transformation defined over rotations and translations.

### B.3. Atomic Action Space

The atomic actions include:

- Horizontal rotation: `turn_right (or left)_deg`

- Vertical rotation: `turn_up (or down) _deg`

- Forward translation: `move_forward _m`

- Vertical translation: `move_up (or down) _m`

The order of actions is based on three stages: (i) adjust the direction to the target position (horizontal / vertical rotation), (ii) move to the target position (forward translation), (iii) adjust the direction to the target camera (horizontal / vertical rotation).

## C. Definition of Reward in ActoR

This section summarizes the reward functions used for reinforcement learning with GRPO, as introduced in Section 3.4. The overall reward is composed of three components: (i) action accuracy, (ii) answer accuracy, and (iii) format correctness.

### C.1. Overview

For each training sample, we compute:

- **Action accuracy reward** $R_{\text{act-acc}} \in [0, 1]$, measuring geometric correctness of predicted action plans;

- **Answer accuracy reward** $R_{\text{ans-acc}} \in [0, 1]$, measuring task-level correctness of the final answer;

- **Format reward** $R_{\text{format}} \in \{0, 1\}$, enforcing the Action-then-Answer output structure.

The final scalar reward used for optimization is a weighted combination of these terms, described in Section 3.4.

### C.2. Action Accuracy Reward ($R_{\text{act-acc}}$)

$R_{\text{act-acc}}$ evaluates whether the predicted action sequence induces a relative camera motion consistent with the ground-truth action plan. It is defined as a smooth pose-level similarity in $[0, 1]$.

Given a predicted action list $\hat{A}$ and a gold action list $A^\star$, we first parse both as JSON. If parsing fails, $R_{\text{act-acc}}$ is set to 0.

Each action list is converted into a relative camera pose $(R, \mathbf{t})$ by composing the rotations and translations specified by the plan. Let $(\hat{R}, \hat{\mathbf{t}})$ and $(R^\star, \mathbf{t}^\star)$ denote the predicted and gold poses, respectively.

We compute translation and rotation errors:

$$d_t = \|\hat{\mathbf{t}} - \mathbf{t}^\star\|_2, \quad d_r = \angle(\hat{R}(R^\star)^\top) \text{ (degrees).} \tag{9}$$

The action accuracy reward is defined as:

$$R_{\text{act-acc}} = \exp\left(-\frac{d_t}{\tau_t}\right) \exp\left(-\frac{d_r}{\tau_r}\right), \tag{10}$$

with fixed temperature parameters $\tau_t$ and $\tau_r$. Optionally, degenerate "stop-like" predicted plans are penalized by setting $R_{\text{act-acc}} = 0$.

### C.3. Answer Accuracy Reward ($R_{\text{ans-acc}}$)

$R_{\text{ans-acc}}$ evaluates the correctness of the final textual answer with respect to the SPAR task definition. The evaluation metric is selected based on the question *sub-category* and *question type* (`select` or `fill`).

Let $\hat{y}$ denote the model prediction and $y^\star$ the gold answer. If either is missing or the required structure cannot be parsed, the reward is set to $0$.

**Selection questions (`select`).** For multiple-choice questions, we use an exact-match criterion with a permissive first-character rule, allowing predictions such as "`C. ...`" to match the gold answer "`C`".

**Numerical fill questions.** For numerical `fill` questions (e.g., distance or depth prediction), we use Mean Relative Accuracy (MRA). Both $\hat{y}$ and $y^\star$ must contain exactly one numeric value; otherwise the reward is $0$. MRA evaluates whether the relative error falls within a set of increasing tolerance intervals and averages the resulting binary accuracies.

**Structured fill questions.** Certain sub-categories require structured outputs and use specialized metrics:

- **Position matching**: Intersection-over-Union (IoU) between predicted and gold bounding boxes.
- **Object-based distance inference**: soft exact match after removing parenthetical annotations.
- **Camera motion inference**: a weighted combination of Gaussian scores over normalized 2D position error and relative depth error.

**View-change inference.** For view-change inference tasks, motion parameters are parsed into a fixed set of canonical dimensions, and MRA is computed independently for each dimension and then averaged.

### C.4. Format Reward ($R_{\text{format}}$)

The format reward $R_{\text{format}}$ enforces strict adherence to the Action-then-Answer protocol. It is a binary reward:

$$R_{\text{format}} \in \{0, 1\}. \tag{11}$$

$R_{\text{format}} = 1$ if and only if all of the following conditions are satisfied:

- The output contains properly closed `<action>` and `<answer>` blocks.
- The contents of `<action>` are parseable as valid JSON.
- The parsed action list covers all required image pairs for the input.

Otherwise, $R_{\text{format}} = 0$.

$R_{\text{format}}$ encourages the model to produce outputs that are directly consumable by downstream parsers and reward functions.

# D. Implementation Details

## D.1. Hyperparameter Settings

This subsection summarizes the hyperparameters used across different training stages. Unless otherwise noted, the same settings are applied to all experiments.

**Learning rate.** We use a fixed learning rate of $1 \times 10^{-5}$ for all training stages, including supervised fine-tuning (SFT) and reinforcement learning.

**Supervised fine-tuning (SFT) and PaStA.** For SFT stages and PaStA, including QA-pair SFT and cold-start SFT for ActoR, we use a batch size of 1 with a single GPU.

**Reinforcement learning (GRPO).** For GRPO training in ActoR, we use:

- Per-GPU batch size: 8

- Number of GPUs: 4 (resulting in a global batch size of 32)

- Number of generated samples per prompt: 8

- KL regularization coefficient: $\beta = 0.01$

**Reward weights.** All reward components are equally weighted during GRPO training, with

$$\lambda_1 = \lambda_2 = \lambda_3 = 1, \tag{12}$$

corresponding to action accuracy, answer accuracy, and format rewards, respectively.

# E. Evaluation Benchmarks

We evaluate our method on a diverse set of spatial reasoning benchmarks, covering both single-image and multi-image settings. These benchmarks are commonly used to assess the geometric reasoning and spatial understanding capabilities of vision–language models.

## E.1. Multi-image Spatial Reasoning

**SPAR-Bench-MV.** SPAR-Bench is a comprehensive visual question–answer benchmark for spatial perception and reasoning, consisting of 20 spatial task categories and 7,211 manually verified QA pairs (Zhang et al., 2025). SPAR-Bench-MV denotes the subset of SPAR-Bench that contains two or more images per sample, focusing on multi-view spatial relations, distance and depth estimation, and camera motion inference across views. The benchmark supports both classification-style accuracy metrics and continuous-valued error-based evaluation.

**MindCube-Tiny.** MindCube-Tiny is a compact multi-view spatial reasoning benchmark derived from the MindCube dataset (Wang et al., 2026). It is designed to probe a model's ability to construct internal spatial mental models from a small number of views. The tasks emphasize perspective transformation, cross-view consistency, and counterfactual spatial reasoning under limited visual observations.

**MMSI-Bench.** MMSI-Bench evaluates multi-image spatial intelligence using 1,000 challenging multiple-choice questions annotated by experts (Yang et al., 2026). The benchmark focuses on real-world multi-view reasoning involving camera–object and inter-object spatial relationships, motion understanding, and region-level attributes. Its carefully designed distractors make the benchmark particularly challenging for current vision–language models.

## E.2. Single-image Spatial Reasoning

**SPAR-Bench-SI.** SPAR-Bench-SI is the single-image subset of SPAR-Bench, consisting of samples with a single image. It includes tasks such as spatial relation recognition, distance estimation, and depth reasoning based solely on monocular visual cues, and serves as a controlled counterpart to SPAR-Bench-MV for analyzing the effect of multi-view information.

**CV-Bench.** CV-Bench is a vision-centric benchmark designed to evaluate spatial reasoning capabilities of vision–language models from a single image (Tong et al., 2024). It covers a range of tasks, including spatial relationship recognition, object counting, depth ordering, and relative distance estimation, with an emphasis on visual grounding rather than language priors.

## F. Limitations

While HATCH demonstrates strong performance on multi-image spatial reasoning, it has several limitations. We discuss two main aspects: (i) failure modes in which HATCH does not improve over baselines, and (ii) the reliance on geometric annotations during training.

### F.1. Failure Modes

We analyze cases where HATCH does not outperform baselines, including the Rotation (Rot.) category in MindCube-Tiny and the Motion and MSR categories in MMSI-Bench. These cases can be grouped into two scenarios.

**Limited spatial overlap across views.** The Rotation category in MindCube-Tiny falls into this case. When images share only a small common region, the supervision signal for PaStA becomes weak, and estimating viewpoint transformations becomes inherently difficult for ActoR. As a result, HATCH does not improve performance and can even underperform the baseline in this setting, since neither cross-view correspondence nor viewpoint-transition actions can be reliably grounded when the shared visual evidence across views is insufficient.

**Temporal inconsistency across views.** Portions of the Motion and MSR categories in MMSI-Bench fall into this case. As discussed in Section 3.1, HATCH assumes a static scene whose geometry is consistent across views. When images are captured at slightly different timestamps, objects or people may move between views, violating this assumption. In such situations, PaStA may produce misleading correspondence signals, as patches can appear to overlap due to object motion rather than true spatial consistency. For example, if an object moves from one patch to another across views, the spatial overlap between these patches may be incorrectly estimated as high, even though they do not correspond to the same physical region. Handling such temporal inconsistencies is an important direction for extending HATCH beyond static scenes, which we leave for future work.

### F.2. Reliance on Geometric Annotations

HATCH relies on camera intrinsics, camera poses, and depth maps during training to construct supervision for PaStA and verifiable rewards for ActoR. Although these signals are never required at inference time, their use during training limits applicability to datasets where such metadata is available (e.g., simulated environments or data captured with depth and pose sensors), and prevents training directly on arbitrary in-the-wild internet images. Nevertheless, we argue that this limitation is mitigated in practice for the following reasons.

**Robustness to moderate annotation noise.** The design of HATCH is inherently tolerant to moderate noise in depth and pose annotations. In ActoR, camera motions are discretized during preprocessing (0.1 m for translation and $10°$ for rotation), so small errors in the underlying poses do not change the resulting supervision. In PaStA, correspondences are computed over coarse patch grids ($n = 4$), making them less sensitive to local geometric perturbations. Together, these choices reduce the precision required of the geometric annotations. We further note that this reliance on geometric signals is not unique to HATCH. Specifically, prior geometry-aware approaches (Wu et al., 2025a; Li et al., 2025) similarly assume access to 3D information during training.

**Improving availability of 3D annotations.** Finally, the availability of 3D annotations is steadily improving with advances in monocular depth and camera pose estimation (Lin et al., 2026), which may further lower the practical barrier to obtaining such signals. Evaluating the effectiveness of training HATCH with automatically estimated geometry is an important direction, but is beyond the scope of this work.

