# OpenReview forum: "From Correspondence to Actions: Human-Like Multi-Image Spatial Reasoning in Multi-modal Large Language Models"
_ICML.cc/2026/Conference — ICML 2026 regular_

### Official Review · Reviewer_4vZk · 2026-03-09

**Soundness:** 3
**Presentation:** 3
**Significance:** 3
**Originality:** 3
**Overall Recommendation:** 3
**Confidence:** 4

**Summary:**

This paper proposes HATCH, a training framework for MLLMs designed to improve multi-image spatial reasoning. Inspired by human cognitive mechanisms, HATCH consists of two main components: (1) Patch-Level Spatial Alignment (PaStA), which uses geometric supervision to align patch representations across different views of the same scene; and (2) Action-then-Answer Reasoning (ActoR), which employs GRPO to train the model to generate explicit viewpoint transition actions before answering the question. The authors evaluate HATCH on three benchmarks and demonstrate that a 3B-parameter model using HATCH outperforms much larger open-weight models and competes with state-of-the-art proprietary models.

**Compliance With Llm Reviewing Policy:**

Affirmed.

**Final Justification:**

My concerns are mostly addressed. However, I still doubt the generalizability of the proposed method in general domains of images.

**Key Questions For Authors:**

Please reply to the weaknesses.

**Limitations:**

No. The limitations of this work should be explained.

**Strengths And Weaknesses:**

**Strengths:**
1. The decomposition of spatial reasoning into "correspondence" (PaStA) and "transformation" (ActoR) is rational. The idea of generating explicit camera actions as an intermediate reasoning step is particularly innovative and adds interpretability.
2. The trained 3B model surpasses Qwen2.5-VL-72B and competing closely with Gemini-3-Pro. This suggests the method is effective.

**Weaknesses:**
1.  PaStA requires camera intrinsics, poses, and depth maps during training. While the authors note these are not needed at inference, this limits the training applicability to datasets where such metadata is available (e.g., simulated environments or captured with sensors), preventing training on arbitrary "in-the-wild" internet images.
2. The action space is fixed to specific atomic operations (turn, move) without parameters (e.g., 30°, 5m). While this enables verifiable rewards, it might restrict the model's ability to reason about more complex or non-rigid spatial transformations not covered by the vocabulary.
3. The experimental results are doubtful: In Section 4.2, the authors claim that HATCH substantially improves upon the base model Qwen2.5-VL-3B across all benchmarks, achieving 53.6% (+17.8 points) on SPAR-Bench-MV, 50.2% (+14.9 points) on MindCube-Tiny. However, these numbers in parentheses are inconsistent with the results in Table 1.

---

> ### Author Rebuttal · Authors · 2026-03-31
>
> Dear Reviewer 4vZk,
>
> Thank you very much for your constructive review and comments. Below, we respond to each question and comment.
>
> # Clarification on training-time dependency on geometric annotations
>
> > W1: PaStA requires camera intrinsics, poses, and depth maps during training…
>
> We agree that HATCH relies on training-time geometric annotations (depth and poses), but this limitation is mitigated in practice for two reasons.
>
> **(1) Data efficiency.**
>
> HATCH achieves strong performance with only 10k samples, compared to much larger datasets used in prior work (e.g., 26k for SpatialLadder and 120k for Spatial-MLLM). We now report the ablation results on training sample size. Performance saturates around 8k samples. This suggests that the required amount of geometric supervision is relatively small.
>
> |  | Low | Middle | High | Avg. |
> | --- | --- | --- | --- | --- |
> | n=2000 | 39.13 | 45.16 | 66.40 | 52.07 |
> | n=4000 | 40.50 | 45.13 | 67.07 | 52.77 |
> | n=6000 | 40.39 | 46.20 | 67.07 | 53.03 |
> | n=8000 | 41.06 | 47.46 | 67.29 | 53.67 |
> | n=10000 | 41.29 | 47.38 | 67.07 | 53.63 |
>
> **(2) Robustness to moderate noise.**
>
> In ActoR, camera motions are discretized (0.1m for moving, 10° for rotating) in preprocessing, so small errors do not affect supervision. In PaStA, correspondences are computed over coarse patch grids (n = 4), making them less sensitive to geometric perturbations.
>
> Together, these design choices make the framework inherently tolerant to moderate noise in depth and pose annotations.
> Additionally, we note that this reliance on geometric signals is not unique to HATCH; prior geometry-aware approaches[1,2] similarly assume access to 3D information during training.
>
> **Improving availability of 3D annotations.**
>
> We also note that the availability of 3D annotations is improving due to advances in monocular depth and pose estimation [3,4], which may further reduce the practical barrier to obtaining such signals. Evaluating the effectiveness of automatically estimated geometry is an important direction, but is beyond the scope of this work.
>
> **Clarification of scope.**
>
> At the same time, we acknowledge that this training assumption should be made clearer, and will explicitly clarify the scope and limitations in the camera-ready version.
>
> [1] Weinzaepfel et al. CroCo: Self-Supervised Pre-training for 3D Vision Tasks by Cross-View Completion. NeurIPS2022.
>
> [2] Wnag et al. Spatial Mental Modeling from Limited Views. ICLR2026.
>
> [3] Yen et al. 3D-PL: Domain Adaptive Depth Estimation with 3D-aware Pseudo-Labeling. ECCV2022.
>
> [4] Zhang et al. StableDepth: Scene-Consistent and Scale-Invariant Monocular Depth. ICCV2025.
>
>
> # Clarification on parameterized action space
>
> > W2: The action space is fixed to specific atomic operations (turn, move) without parameters (e.g., 30°, 5m)...
>
> We would like to clarify that the action space does include continuous parameters.
> As shown in Figure 2 (Generated Response in the ActoR module) and Figure 5, the generated actions include parameters such as rotation angles (e.g., 30°) and movement distances (e.g., 5m). Therefore, the action space is not limited to a fixed discrete set, and can express a wide range of spatial transformations.
>
> We will revise the description of the reasoning formulation (L222–242) to make this clearer.
>
> # Correction of reported performance gains
>
> > W3: In Section 4.2, the authors claim that HATCH substantially improves upon the base model... However, these numbers in parentheses are inconsistent with the results in Table 1.
>
> Thank you for pointing this out.
>
> This is a mistake in our writing, and we apologize for the confusion.
> The values in Table 1 are correct, but the improvement numbers stated in Section 4.2 contain a writing error. Based on Table 1, the correct improvements over Qwen2.5-VL-3B are: 53.6% (+28.7 points) on SPAR-Bench-MV, and 50.2% (+12.4 points) on MindCube-Tiny.
>
> We will correct this in the camera-ready version.
>
> # Planned additions to the Limitations section
>
> > Limitations: No. The limitations of this work should be explained.
>
> We agree that the limitations of our method should be more clearly discussed.
>
> In the camera-ready version, we will explicitly include a Limitations section, covering aspects such as the reliance on training-time geometric annotations as discussed above.

---

> > ### Author Rebuttal · Reviewer_4vZk · 2026-04-02
> >
> > Thanks for your response. My concerns are mostly addressed. However, I still doubt the generalizability of the proposed method in general domains of images.

---

> > > ### Author Response · Authors · 2026-04-02
> > >
> > > Dear Reviewer 4vZk,
> > >
> > > Thank you for your follow-up. We are glad that most of your concerns have been addressed.
> > > We interpret your follow-up question as focusing on the training-time dependency on geometric annotations. We further clarify this point below.
> > >
> > > # Additional clarification on training-time dependency on geometric annotations
> > >
> > > As mentioned in our initial response, we believe that the dependency on geometric annotations is mitigated in practice due to (1) strong data efficiency and (2) robustness to moderate noise in the annotations.
> > >
> > > Beyond these points, we would like to clarify how this design choice should be interpreted in the current research landscape: **recent multi-image spatial reasoning methods commonly leverage geometry-based supervision at training time**. For example, SpatialLadder [1; ICLR2026] explicitly leverages 3D reconstructions from ScanNet to construct training data, and similar use of geometric signals is also adopted in Spatial-MLLM [2; NeurIPS2025] and MindCube [3; ICLR2026].
> > >
> > > Within this paradigm, HATCH is particularly efficient and practical. Compared to prior methods that require substantially larger datasets (e.g., 26k samples for SpatialLadder and 120k for Spatial-MLLM), HATCH achieves stronger performance with only 10k samples. This significantly reduces the amount of geometric supervision required, making the approach more accessible in realistic scenarios.
> > >
> > > Therefore, **rather than introducing an additional constraint, HATCH can be viewed as improving the practicality of geometry-aware training by minimizing the required supervision while achieving state-of-the-art performance**. This makes it a more scalable step toward applying spatial reasoning models beyond curated 3D datasets.
> > >
> > > We will clarify this perspective more explicitly in the camera-ready version.
> > >
> > > [1] Li et al. SpatialLadder: Progressive Training for Spatial Reasoning in Vision-Language Models. ICLR2026.
> > >
> > > [2] Wu et al. Spatial-MLLM: Boosting MLLM Capabilities in Visual-based Spatial Intelligence. NeurIPS2025.
> > >
> > > [3] Wang et al. MindCube: Spatial Mental Modeling from Limited Views. ICLR2026.
> > >
> > > # General image (and video) capabilities beyond spatial reasoning
> > >
> > > If your concern instead relates to whether HATCH preserves general image understanding capabilities beyond spatial reasoning, we provide additional results below.
> > >
> > > We evaluate general image (and video) understanding by comparing Qwen2.5-VL-3B (base), SpatialLadder, and HATCH on standard benchmarks (MMMU, MMMU-Pro, MMBench-EN, and Video-MME):
> > >
> > > |  | **MMMU** | **MMMU-Pro** | **MMBench-EN** | **Video-MME** |
> > > | --- | --- | --- | --- | --- |
> > > | Qwen2.5-VL-3B | 47.0 | 31.5 | 78.8 | 57.6 |
> > > | SpatialLadder | 45.6 | 24.1 | 73.6 | 52.3 |
> > > | HATCH | 46.3 | 29.0 | 75.7 | 56.1 |
> > >
> > > HATCH maintains performance close to the base model on MMMU (46.3 vs 47.0) and Video-MME (56.1 vs 57.6), indicating strong preservation of general capabilities. While we observe moderate drops on MMMU-Pro and MMBench-EN, HATCH consistently outperforms SpatialLadder across all benchmarks, suggesting better retention of general-purpose abilities compared to prior spatial reasoning approaches.
> > >
> > > These results indicate that HATCH generalizes well beyond spatial reasoning while maintaining competitive performance in general domains.

---

### Official Review · Reviewer_pqMX · 2026-03-10

**Soundness:** 3
**Presentation:** 3
**Significance:** 3
**Originality:** 3
**Overall Recommendation:** 3
**Confidence:** 3

**Summary:**

This paper proposes HATCH, a training framework inspired by human spatial cognition for improving multi-image spatial reasoning in MLLMs. It consists of two stages: PaStA, which aligns patch-level geometry across views using depth and camera poses, and ActoR, which requires the model to generate viewpoint transformation actions before answering and optimizes this process with verifiable rewards via GRPO. Experiments show improved spatial reasoning performance on both multi-image and single-image settings at the 3B scale.

**Compliance With Llm Reviewing Policy:**

Affirmed.

**Final Justification:**

After reading the rebuttal, my major concerns have been addressed. However, I am still concerned about the method’s reliance on the static-scene assumption and its generalizability. Therefore, I maintain my original rating.

**Key Questions For Authors:**

- Does this training paradigm preserve general image and video understanding ability beyond spatial reasoning?

- PaStA assumes a static scene for geometric projection. How robust is it when objects or people move across views?


If the authors can provide additional experiments or convincingly address these concerns in the rebuttal, I would like to increase my score.

**Limitations:**

yes

**Strengths And Weaknesses:**

Strengths:

- The method is conceptually clean and well motivated, translating cross-view correspondence and stepwise viewpoint transformation into explicit training objectives.

- The two-stage design is reasonable, with PaStA focusing on geometric alignment and ActoR improving reasoning.

- The paper is clearly written and easy to follow.

Weaknesses:

- The framework depends heavily on accurate 3D annotations, including depth and camera poses, which may limit scalability to in-the-wild data.

- ActoR uses autoregressive action generation, which may suffer from error accumulation if early actions are wrong.

---

> ### Author Rebuttal · Authors · 2026-03-31
>
> Dear Reviewer pqMX,
>
> Thank you very much for your insightful review and comments. Below, we respond to each question and comment.
>
> # Clarification on training-time dependency on geometric annotations
>
> > W1: The framework depends heavily on accurate 3D annotations…
>
> We agree that our method relies on training-time geometric annotations, but this limitation is mitigated by its data efficiency and robustness to moderate noise.
>
> Please refer to the "Clarification on training-time dependency on geometric annotations" section in our response to Reviewer 4vZk for details.
>
> # Analysis of error accumulation in ActoR
>
> > W2: ActoR uses autoregressive action generation, which may suffer from error accumulation if early actions are wrong.
>
> We agree that, in principle, autoregressive action generation can suffer from error accumulation if early actions are incorrect.
> However, we consider this issue has limited practical impact in our setting for the following reasons.
>
> **(1) Actions act as reasoning scaffolds.**
>
> The generated actions are not executed externally but serve as intermediate reasoning steps. As a result, small inaccuracies in early actions do not necessarily propagate to incorrect final answers. This is consistent with prior findings in reasoning literature, where intermediate reasoning steps can be imperfect yet still lead to correct final predictions, as long as they provide useful structural guidance[1].
>
> **(2) Empirical evidence.**
>
> Removing ActoR (“w/o ActoR”) consistently degrades performance (Table 2), and ActoR outperforms natural language chain-of-thought reasoning under the same setup (Table 3), showing that structured action reasoning remains effective in practice even if some degree of error accumulation occurs.
>
> **Future directions.**
> We agree that mitigating error accumulation is an interesting direction for future work.
> One promising extension is to allow corrective actions rather than strictly sequential ones. Such a design could enable the model to recover from early mistakes by refining later steps, potentially improving robustness.
>
> [1] Paul et al.  Making Reasoning Matter: Measuring and Improving Faithfulness of Chain-of-Thought Reasoning. Findings of EMNLP2024.
>
> # General image and video capabilities beyond spatial reasoning
>
> > Q1: Does this training paradigm preserve general image and video understanding ability beyond spatial reasoning?
>
> We evaluate general image and video understanding by comparing Qwen2.5-VL-3B (base), SpatialLadder (a strong baseline for spatial reasoning), and HATCH on standard benchmarks (MMMU, MMMU-Pro: STEM capability with image inputs, MMBench-EN: image understanding, Video-MME: video understanding):
>
> |  | MMMU | MMMU-Pro | MMBench-EN| Video-MME |
> | --- | --- | --- | --- | --- |
> | Qwen2.5-VL-3B | 47.0 | 31.5 | 78.8 | 57.6 |
> | SpatialLadder | 45.6 | 24.1 | 73.6 | 52.3 |
> | HATCH | 46.3 | 29.0 | 75.7 | 56.1 |
>
> HATCH maintains performance close to the base model on MMMU and Video-MME, indicating strong preservation of general capabilities. While we observe moderate drops on MMMU-Pro and MMBench-EN, HATCH consistently outperforms SpatialLadder across all benchmarks, suggesting better retention of general-purpose abilities compared to prior spatial reasoning approaches.
>
> Overall, these results indicate that HATCH largely preserves general capabilities while improving spatial reasoning.
>
> # Robustness under dynamic scenes
>
> > Q2: PaStA assumes a static scene for geometric projection. How robust is it when objects or people move across views?
>
> We acknowledge that PaStA assumes a static scene and is not designed to handle significant object or human motion across views. Consistent with this limitation, we observe relatively weaker performance in categories involving motion, such as Motion and MSR in MMSI-Bench, where temporal discrepancies across views are more prominent.
>
> At the same time, our primary focus is on static multi-view scenes, and handling temporal inconsistencies is outside the scope of this work. This setting is practically important and widely applicable, for example, in scenarios involving multiple surveillance cameras or multi-robot systems observing the same environment from different viewpoints. Moreover, as shown in Table 1, even in largely static benchmarks such as SPAR-Bench-MV, existing models still struggle significantly (e.g., 39.9 for GPT-4.1, 35.4 for Qwen2.5-VL-72B, and 24.9 for Qwen2.5-VL-3B), with some results close to chance level. This suggests that, **while handling motion is an important direction, improving spatial reasoning in static multi-view settings remains a meaningful and challenging problem.**
>
> We agree that clarifying this assumption is important. In the camera-ready version, we will explicitly state the focus on static scenes in the Introduction and discuss temporal discrepancies across views as a limitation.

---

> > ### Author Rebuttal · Reviewer_pqMX · 2026-04-03
> >
> > I thank the authors for their response. My main concerns have been addressed. However, I still remain concerned about the reliance on the static-scene assumption and the generalizability of the method.

---

> > > ### Author Response · Authors · 2026-04-04
> > >
> > > Dear Reviewer pqMX,
> > >
> > > Thank you for your follow-up. We are glad that our previous responses addressed most of your concerns.
> > >
> > > Below, we further clarify the points regarding the static-scene assumption and generalizability.
> > >
> > > # Clarification on the static-scene assumption
> > >
> > > ---
> > >
> > > We agree that our method assumes static scenes and does not handle significant object or human motion across views. This limitation is reflected in our results, where gains on motion-related subsets are smaller. However, we believe this should be understood as a scoped limitation rather than a flaw that undermines the main contribution.
> > >
> > > First of all, **static multi-view spatial reasoning is itself a challenging open problem**: even frontier models remain far from strong performance in this regime (e.g., 35.4 for Qwen2.5-VL-72B on SPAR-Bench-MV).
> > > **Our contribution is to improve performance on this core bottleneck**, which we believe is a meaningful step toward broader multi-image spatial reasoning, rather than an overly narrow simplification of the problem.
> > >
> > > In addition, static multi-view reasoning has standalone practical value in settings such as multi-camera robotic perception [1] and multi-agent (e.g., drone) coordination [2], where the scene is often approximately consistent across views.
> > >
> > > While handling dynamic scenes is beyond the scope of this work, we note that there are several natural directions to extend our framework. One possible direction is to extend patch-level spatial correspondence in PaStA to object-level correspondence, by aligning representations of the same object across views even under motion.
> > >
> > > We will clarify this scope in the camera-ready version by explicitly stating the static-scene assumption, discussing the limitation more directly, and positioning dynamic-scene reasoning as an important future extension.
> > >
> > >
> > > # Clarification on generalizability
> > >
> > > ---
> > >
> > > We would like to clarify that our claim of generalizability is modest and supported in two specific senses.
> > >
> > > **(1) Transfer beyond the target setting.**
> > >
> > > As already shown in Table 4 in the submission, HATCH improves not only multi-image spatial reasoning but also single-image spatial reasoning. As described in the previous response, it also preserves general image and video understanding performance on standard benchmarks (MMMU, MMMU-Pro, MMBench-EN, Video-MME), while outperforming prior spatial reasoning methods such as SpatialLadder. This suggests that the proposed training does not overfit to the target setting or degrade broader visual understanding ability.
> > >
> > > **(2) Limited dependence on idealized supervision.**
> > >
> > > As discussed previously, our reliance on geometric supervision is mitigated by strong data efficiency and robustness to moderate noise (at least 0.1 m for translation and 10° for rotation). Moreover, geometry-based supervision is also used in recent prior work on spatial reasoning (e.g., SpatialLadder [3; ICLR2026], Spatial-MLLM [4; NeurIPS2025], MindCube [5; ICLR2026]). In this context, HATCH improves practicality by achieving strong performance with substantially fewer training samples, rather than requiring large amounts of carefully curated supervision.
> > >
> > > [1] Jang et al. Multi-Camera-Based Human Activity Recognition for Human–Robot Collaboration in Construction. Sensors 2023, 23(15).
> > >
> > > [2] Pedroche et al. Drone Swarm for Distributed Video Surveillance of Roads and Car Tracking. Drones 2024, 8(11).
> > >
> > > [3] Li et al. SpatialLadder: Progressive Training for Spatial Reasoning in Vision-Language Models. ICLR2026.
> > >
> > > [4] Wu et al. Spatial-MLLM: Boosting MLLM Capabilities in Visual-based Spatial Intelligence. NeurIPS2025.
> > >
> > > [5] Wang et al. MindCube: Spatial Mental Modeling from Limited Views. ICLR2026.

---

### Official Review · Reviewer_KuGu · 2026-03-12

**Soundness:** 3
**Presentation:** 3
**Significance:** 3
**Originality:** 3
**Overall Recommendation:** 5
**Confidence:** 3

**Summary:**

This work introduces Hatch, a finetuning framework that explicitly incorporates cross-view correspondence and viewpoint transformation into MLLM multi-image spatial reasoning. The method design is motivated not only from cognitive theory but also from prior machine learning models, making the overall statement solid. Extensive experiments demonstrate the effectiveness of the method.

**Compliance With Llm Reviewing Policy:**

Affirmed.

**Final Justification:**

The author fully addressed my concerns and I believe this is a valuable work.

**Key Questions For Authors:**

1. In the dataset, does the action include multiple image correspondence or just two images?

**Limitations:**

See Weaknesses.

**Strengths And Weaknesses:**

Strengths:

1. The paper is well-motivated from two aspects: prior machine learning framework and cognitive science.

2. The proposed method indeed thrives in most of the cases for multi-image spatial reasoning.

3. The paper is well-written and easy to follow.

Weaknesses:

1. It is unclear how the result would be if only direct SFT is applied to the model.

2. Lack of analysis on results where the proposed method cannot achieve better results than base model or other methods, e.g. Rot. category for MindCube-Tiny in Table 1.

3. It is unclear whether the performance of the finetuned model will be better if the scale of the data used at finetuning increases.

---

> ### Author Rebuttal · Authors · 2026-03-31
>
> Dear Reviewer KuGu,
>
> Thank you very much for your constructive review and positive feedback. Below, we respond to each question and comment.
>
> # Clarification of comparison with direct SFT
>
> > W1: It is unclear how the result would be if only direct SFT is applied to the model.
>
> As described in L378–380, this comparison is already included in Table 2 as the “w/o both” setting, which corresponds to direct SFT without PaStA or ActoR. As shown in Table 2, HATCH consistently outperforms this direct SFT baseline, demonstrating the benefit of explicitly modeling both cross-view correspondence and viewpoint transitions.
>
> # Analysis of challenging cases for HATCH
>
> > W2: Lack of analysis on results where the proposed method cannot achieve better results than base model or other methods, e.g. Rot. category for MindCube-Tiny in Table 1.
>
> We analyze cases where HATCH does not outperform baselines, including the Rotation (Rot.) category in MindCube-Tiny and Motion / MSR categories in MMSI-Bench. These can be grouped into two main scenarios:
>
> 1. Limited spatial overlap across views: The Rot. category in MindCube-Tiny falls into this case. When images share only a small common region, the supervision signal for PaStA becomes weak, and estimating viewpoint transformations becomes inherently difficult for ActoR in principle. As a result, HATCH does not improve performance and can even underperform the baseline in this setting.
>
> 2. Temporal inconsistency across views (e.g., moving objects): Portions of the Motion and MSR categories in MMSI-Bench fall into this case. When images are captured at slightly different timestamps, objects or people may move across views. In such situations, PaStA may produce misleading correspondence signals, as patches can appear to overlap due to object motion rather than true spatial consistency. For example, if an object moves from patch A to patch B, the spatial overlap between these patches may be incorrectly estimated as high, even though they do not correspond to the same physical region.
>
> We will add a discussion of these failure modes in the Limitations section of the camera-ready version.
>
> Beyond the current scope, we believe these challenges could be addressed by incorporating world modeling capabilities (e.g., inferring occluded regions or modeling object dynamics)[1,2], which may improve robustness in both low-overlap and dynamic scenarios.
>
> [1] Yang et al. MindJourney: Test-Time Scaling with World Models for Spatial Reasoning. NerIPS2025.
>
> [2] Yu et al. When and How Much to Imagine: Adaptive Test-Time Scaling with World Models for Visual Spatial Reasoning. arXiv2026.
>
> # Ablation study on training data size
>
> > W3: It is unclear whether the performance of the finetuned model will be better if the scale of the data used at finetuning increases.
>
> We conducted an ablation study on the number of training samples on SPAR-Bench-MV.
>
> |  | Low | Middle | High | Avg. |
> | --- | --- | --- | --- | --- |
> | n=2000 | 39.13 | 45.16 | 66.40 | 52.07 |
> | n=4000 | 40.50 | 45.13 | 67.07 | 52.77 |
> | n=6000 | 40.39 | 46.20 | 67.07 | 53.03 |
> | n=8000 | 41.06 | 47.46 | 67.29 | 53.67 |
> | n=10000 | 41.29 | 47.38 | 67.07 | 53.63 |
>
> As shown, performance improves steadily with more data and begins to saturate around 8k–10k samples. This suggests that HATCH scales well with data, while already achieving strong performance in a relatively low-data regime.
>
> Importantly, our method uses only 10k samples, whereas prior baselines are trained on substantially larger datasets (e.g., 26k [3] and 120k [4]). This indicates that HATCH is relatively data-efficient, benefiting from structured supervision rather than relying purely on scale.
>
> We note that exploring larger-scale training (e.g., 20k+ samples) is an interesting direction; however, due to computational constraints, we leave this as future work.
>
> [3] Li et al. SpatialLadder: Progressive Training for Spatial Reasoning in Vision-Language Models. ICLR2026.
>
> [4] Wu et al. Spatial-MLLM: Boosting MLLM Capabilities in Visual-based Spatial Intelligence. NeurIPS2025.
>
> # Clarification of action formulation for multi-image relationships
>
> > Q1: In the dataset, does the action include multiple image correspondence or just two images?
>
> The action annotations include multiple image correspondences. As described in L231-232, we generate actions for all unordered image pairs (i, j) with i < j.  For example, given three images (A, B, and C), the model generates action sequences for all pairs: (A, B), (A, C), and (B, C). These pairwise actions collectively capture multi-image relationships within each sample.

---

> > ### Author Rebuttal · Reviewer_KuGu · 2026-04-02
> >
> > I would like to thank the author for their point-by-point response, they resolved my concerns. I would raise my score from 4 to 5 as I believe this work is insightful and my concerns are fully addressed.

---

### Official Review · Reviewer_Ywjy · 2026-03-13

**Soundness:** 3
**Presentation:** 3
**Significance:** 3
**Originality:** 3
**Overall Recommendation:** 5
**Confidence:** 3

**Summary:**

The paper studies how to make multimodal large language models (MLLMs) reason across multiple views of the same scene, rather than single-image spatial reasoning and proposes a training framework, HATCH, to supervise both cross-view correspondence and step-wise viewpoint transformation explicitly. They also propose Patch-Level Spatial Alignment (PaStA) to use training-time geometry signals to align patch features across different views, and Action-then-Answer Reasoning (ActoR) to train the model to output explicit camera-transition actions before giving the answer.

**Compliance With Llm Reviewing Policy:**

Affirmed.

**Final Justification:**

I thank the authors for their response. The rebuttal addresses all my concerns and I hope that the authors will include them in the final revision. After reading other reviewers' comments and the authors' responses, I have decided to stay with my initial score.

**Key Questions For Authors:**

Please refer to the weaknesses above.

**Limitations:**

yes

**Strengths And Weaknesses:**

Strengths:
The paper has an interesting core idea, and the decomposition into "how to look" and "how to move" is intuitive.
Also, the proposed supervision is more explicit than most prior work, and the action-based intermediate reasoning is more verifiable. The authors also provide extensive experiments in the paper.

Weakness:
I am curious to know how sensitive the method is to noisy or approximate geometry. The paper acknowledges that MMSI remains difficult, but I am curious about what variations of the questions are harder for the method.

---

> ### Author Rebuttal · Authors · 2026-03-31
>
> Dear Reviewer Ywjy,
>
> Thank you very much for your constructive review and positive feedback. Below, we respond to each question and comment.
>
> # Sensitivity to noise in ground-truth geometry annotation
>
> > W1-1: I am curious to know how sensitive the method is to noisy or approximate geometry.
>
> Our method shows robustness to moderate noise in geometric annotations, particularly within ranges of approximately 0.1 meters in translation and 10 degrees in rotation.
>
> Importantly, our training setup itself already allows for such levels of noise:
>
> (1) In ActoR, ground-truth camera operations are discretized during preprocessing by rounding translations to 0.1 meters and rotations to 10 degrees in the preprocessing stage. As a result, the supervision signal does not distinguish variations within these ranges.
>
> (2) For PaStA, correspondences are computed over a coarse patch grid (n × n with n = 4 in Eq. 5). Each patch covers a relatively large spatial region, which makes the correspondence targets less sensitive to small geometric perturbations.
>
> Given that both components operate under this level of approximation, our experimental results indicate that the method performs well despite such imperfect geometry.
> Overall, this suggests that HATCH is robust in practice to moderate inaccuracies in depth and pose annotations.
> We will clarify this robustness to moderate geometric noise in the revised manuscript (e.g., at the end of the Methodology section).
>
> # Analysis of challenging cases for HATCH
>
> > W1-2: The paper acknowledges that MMSI remains difficult, but I am curious about what variations of the questions are harder for the method.
>
> We identify two types of scenarios that are particularly challenging for our method:
>
> 1. Limited spatial overlap across views: When images share only a small common region, the supervision signal for PaStA becomes weak, and estimating viewpoint transformations becomes inherently difficult for ActoR in principle. This is reflected in the Rotation (Rot.) category of MindCube-Tiny, where HATCH underperforms the baseline.
> 2. Temporal inconsistency across views (e.g., moving objects): When images are captured at slightly different timestamps, objects or people may move across views. In such cases, PaStA can produce misleading correspondence signals, as patches may appear to overlap due to object motion rather than true spatial consistency. For example, if an object moves from patch A to patch B, the spatial overlap between these patches may be incorrectly estimated as high, even though they do not correspond to the same physical regions. This issue is observed in portions of the Motion and MSR categories in MMSI-Bench, where HATCH again underperforms the baseline.
>
> We will add a discussion of these failure modes in the Limitations section of the camera-ready version.
>
> Beyond the current scope, we believe these challenges could be addressed by incorporating world modeling capabilities (e.g., inferring occluded regions or modeling object dynamics)[1,2], which may improve robustness in both low-overlap and dynamic scenarios.
>
> [1] Yang et al. MindJourney: Test-Time Scaling with World Models for Spatial Reasoning. NerIPS2025.
>
> [2] Yu et al. When and How Much to Imagine: Adaptive Test-Time Scaling with World Models for Visual Spatial Reasoning. arXiv2026.

---

> > ### Author Rebuttal · Reviewer_Ywjy · 2026-04-01
> >
> > I appreciate the authors' responses and acknowledge that my concerns have been addressed.

---

### Decision · Program_Chairs · 2026-04-30

**Decision:**

Accept (regular)

**Comment:**

This paper introduces HATCH, a training framework designed to improve multi-image spatial reasoning in MLLMs. It consists of two main components: Patch-Level Spatial Alignment and Action-then-Answer Reasoning. Experiments on three benchmarks demonstrate improved spatial reasoning performance in both multi-image and single-image settings.

During the review and discussion phase, the authors have made substantial efforts to address many of the questions and concerns raised. While some issues remain, such as the method’s reliance on a static-scene assumption and the need for a more systematic evaluation of its generalizability to broader image and video domains (e.g., OCR benchmarks like OCRBench and DocVQA, 2D/3D grounding such as RefCOCO-avg, etc), these do not undermine the overall contribution. I also see the potential value of this work in applications such as multi-camera surveillance and multi-agent coordination. Given these considerations, I recommend acceptance.